# How is hygiene behaviour affected by conflict and displacement? A qualitative case study in Northern Iraq

Sian White[1]*, Thomas Heath[2], Waleed Khalid Ibrahim[3], Dilveen Ihsan[4], Karl Blanchet[5], Val Curtis[1], Robert Dreibelbis[1]

1 Department of Disease Control, London School of Hygiene and Tropical Medicine, London, United Kingdom, 2 Operations Department, Action Contre la Faim, Paris, France, 3 Independent Researcher, Nutrition and Dietetics Department, College of Medical Technology, Cihan University, Erbil, Kurdistan Region of Iraq, 4 Independent Researcher, Dohuk, Kurdistan Region of Iraq, 5 Geneva Centre of Humanitarian Studies, Université de Genève, Geneva, Switzerland

* sian.white@lshtm.ac.uk

**Data Availability Statement:** Some relevant data are within the Supporting Information files. The transcripts of the interviews and group discussions are publicly available in redacted form via an open repository which can be accessed via the following

## Abstract

This research aimed to qualitatively explore whether the determinants of handwashing behaviour change according to the duration of displacement or the type of setting that people are displaced to. We conducted an exploratory qualitative study in three different post-conflict settings in Northern Iraq–a long-term displacement camp, a short-term displacement camp, and villages where people were returning to post the conflict. We identified 33 determinants of handwashing in these settings and, of these, 21 appeared to be altered by the conflict and displacement. Determinants of handwashing behaviour in the post-conflict period were predominantly explained by disruptions to the physical, psychological, social and economic circumstances of displaced populations. Future hygiene programmes in post-conflict displacement settings should adopt a holistic way of assessing determinants and design programmes which promote agency, build on adaptive norms, create an enabling environment and which are integrated with other aspects of humanitarian response.

## Introduction

Conflicts often create the 'perfect storm' of circumstances to enable communicable disease transmission [1]. This is because in the wake of conflict infrastructure and water and sanitation systems are often damaged, populations are displaced to densely populated areas, markets collapse, and health facilities are weakened or overburdened [2]. Consequently diarrhoeal and respiratory infections are the leading cause of preventable illness and death during crises [3]. Handwashing with soap has the potential to reduce the burden of diarrhoeal diseases, respiratory diseases and other outbreak-related pathogens [4–6]. However, handwashing rates are low globally [7] and likely to be even lower in post-conflict displacement contexts.

Behaviour change theorists suggest that for behaviour change programmes to be effective, they must address the determinants that influence the behavioural outcome [8–11]. A recent systematic review of the determinants of handwashing behaviour found that the quality of

citation: White, Sian (2021): Interviews and group discussions with displaced populations in Iraq on the determinants of handwashing behaviour. figshare. Dataset. https://doi.org/10.6084/m9.figshare.17263829. Observational data, visual data and data from handwashing demonstrations is not available because it is not possible to make this data anonymous.

**Funding:** Funding for this research was provided by United States Agency for international development's Bureau of Humanitarian Assistance via grant number AID-OFDA-G-16-00270. The funders had no role in study design, data collection and analysis, decision to publish, or preparation of the manuscript. This grant contributed to the salaries of Sian White, Thomas Heath, Waleed Khalid Ibrahum and Dilveen Ihsan, but did not fully fund these.

**Competing interests:** The authors have declared that no competing interests exist.

studies on hygiene behavioural determinants remained poor and that studies disproportionally reported on personal characteristics and cognitive determinants [12]. Determinants such as routines, norms, contextual factors, motives, and the physical and biological environments were less frequently described in the literature. Although the review conducted a sub-analysis about the determinants of handwashing behaviour during humanitarian crises, no conclusions could be drawn due to the limited number of studies in these settings. Other reviews have also highlighted the lack of hygiene behaviour change research specific to crisis-affected settings, the poor quality of this research and the challenges of doing handwashing behaviour change in these settings [4,13,14]. However, broader literature indicates that major life events, and changes to physical and social circumstances, are likely to interrupt prior habits, create new norms, and introduce new enablers or barriers to behaviour [15,16], meaning that behavioural changes are likely to occur during crises even if poorly understood to date. Understanding the determinants of handwashing behaviour during crises has therefore been identified as a sector priority [17,18], particularly as humanitarians are under increasing pressure to develop guidelines and programmes that are based on evidence-based [19,20].

Most studies on handwashing behaviour in crisis-affected settings have used survey-based methods to understand determinants and self-reported behaviour [21–25]. Survey-based approaches can only explore determinants that they identify in advance and these typically focus on knowledge, risk perception, personal characteristics, and capability. Self-reported handwashing behaviour measures are also known to over-estimate actual practice [26,27]. Given that the determinants of hygiene are a poorly understood phenomenon, rooted in human experience, and often driven by sub-conscious factors, qualitative methods may be better placed to facilitate meaning making on this topic.

This research aims to qualitatively explore whether the determinants of handwashing behaviour change according to the type of setting that people are displaced to, and the stage of their displacement.

## Methods

### Study sites

This study took place in three study sites in Northern Iraq between June and August 2017 during the peak of the offensive against the Islamic State of Iraq and the Levant (hereafter referred to as *Da'ish*).

We selected research sites purposively to reflect different durations of displacement (e.g. a short-term displacement camp, a long-term displacement camp and returnee villages), different social and physical settings within a conflict (e.g. comparing 'closed' verses 'open' camp settings, and comparing tented shelters to damaged buildings), and differences in water, sanitation, and hygiene (WASH) coverage. The first site was the tented Nargizliya Camp located within Dohuk Governorate in the Kurdistan Region of Iraq. Founded 6 months prior to data collection, Nargizliya housed 9,905 people who were predominantly Arab and had fled from the city of Mosul and its surrounding villages. As a 'closed camp', residents in Nargizlyia were not allowed to leave without permission, and access to communications (e.g. mobile phones) was not permitted. The second site was Sheikhan Camp, another tented camp in Dohuk Governorate. Sheikhan Camp held a population of 5,371 Yazidi (Êzidî) people who had fled from the city of Sinjar and its surrounding villages in summer 2014 and who had resided in the camp for three years. Residents in Sheikhan were able to come and go from the camp freely and many worked in the nearby town. The third site included two neighbouring villages on the outskirts of Mosul in the Ninewa Governorate of Iraq. Residents of these villages had been displaced during the conflict and had returned within the last few months to homes damaged

**Table 1. WASH characteristics informing the selection of the three research sites.**

| WASH activities | Nargizlyia Camp | Sheikhan Camp | Villages |
|---|---|---|---|
| **Toilet facilities** | Shared between 6 tents, built by NGOs* | Private, built by NGOs | Private, built by households |
| **Bathing spaces** | Shared between 6 tents | Private built by NGOs | Private, built by households |
| **Kitchen facilities** | Shared between 6 tents, built by NGOs | Private, built by NGOs | Private, built by households |
| **Water supply** | Tank at the shared WASH facilities | Tank at the shared WASH facilities | Water trucking or piped water |
| **Hygiene products** | Distributed by NGOs | Purchased by households | Purchased by households, irregular access to markets |

*NGOs = Non-government organisations.

during the conflict. At the time of the research, 134 Arab or Shabak families had returned to these villages. The villages were also home 30 additional families who were ethnically similar Internally Displaced Persons (IDPs) from neighbouring villages. Displaced families either shared homes with residents of the village or lived in damaged buildings which others had not returned to. Table 1 summarises the WASH characteristics of all sites.

## Research framework

This research used observations, focus group discussions, in-depth interviews and handwashing demonstrations to explore behavioural determinants. The research was informed by Behaviour Centred Design (BCD) [9] which draws on evolutionary and environmental psychology to define domains of behaviour including cognitive processes, socio-demographic characteristics, the settings where behaviours take place (and the infrastructure, objects, norms, roles and routines that are associated with these settings) and the physical, social and contextual environment. In total, 16 categories of determinants were pre-identified for exploration [28]. S1 Appendix defines each of the BCD determinants in relation to handwashing and indicates which methods were used to explore them. The group discussions and interviews were designed to explore current handwashing behaviours in each study site and perspectives on what shaped behaviours pre and post the conflict. Observations and demonstrations helped to understand current behaviour in context.

## Data collection methods

**Observations.** Unstructured observations took place in 20 households across the three sites and were designed to understand behaviour within a contextual setting. Observations were scheduled for 3 hours, typically beginning at 8am (depending on local security). Observers wrote down all actions done by household members and the time actions took place. Observers recorded the availability of soap and participant behaviour during 'critical moments' for handwashing which were defined as handwashing after using the toilet, or cleaning a child's bottom, and before preparing food, eating food or feeding a child. Households participating in the observation were informed that we were interested in learning about 'daily routines' and were therefore not aware that the study was specifically interested in handwashing, so to reduce reactivity. Given that we were taking notes on all action that occurred, it was also unlikely that participants could deduce that we were interested in handwashing behaviour. To monitor the quality of the data recorded, two households were observed at the same time (by two of the authors—DI and WKI) while the lead researcher (SW) moved between households and took notes to allow cross-checking and consistency.

**Focus group discussions.** Ten group discussions were completed across the study sites involving 93 people. Four participatory activities were included in the group discussions to

explore current and past hygiene challenges, priorities, perceived risk and preferences related to infrastructure and soap. Participatory activities included free listing and categorisation of priorities, risk scaling and appraising the characteristics handwashing products and infrastructure. See S2 Appendix for detailed descriptions of the participatory activities. Group discussions were 1.5 hours in duration.

**In depth interviews.** A total of 98 interviews were completed across study sites. A total of 8 participatory activities were used within the interviews to explore current and past hygiene challenges, water use, roles, routines, norms, motives, social networks, and contextual determinants. Participatory activities included the ranking of hygiene challenges, eliciting responses to scenarios about water use, routine scripting, predictions about the normative behaviour of others, social network mapping and drawing exercises to understand experiences of the conflict and displacement. S3 Appendix provides a detailed description of the participatory activities undertaken within interviews. Interviews were 45 minutes in duration.

**Handwashing demonstrations.** Handwashing demonstrations were conducted with 24 individuals. Participants demonstrated how they normally wash their hands after using the toilet. This method typically reflects the participant's 'ideal practice' (since they are aware of the observer and the target behaviour), but can be useful for understanding barriers and enablers within the behavioural setting [29].

## Sampling

Participants were selected purposively to reflect a diversity of age, gender, geographies, ethnicity or religion, and access to resources. Local maps were reviewed with camp managers or village leaders and used to identify households. Sampling continued until a degree of saturation was met for each method. The number of people involved in each method is summarised in Table 2 and explained in more detail in S2 and S3 Appendices.

## Data collection, management, and analysis

Interviews, group discussions and observations were conducted by the three authors (SW, DI and WKI). We were a team with mixed cultural backgrounds (SW is British and DI and WKI are Kurdish) and all had prior experience with qualitative data collection. SW provided two days of classroom-based training on the research methods and we spent two days piloting the methods in a similar setting to ensure all research team members understood the methods and were able to apply them in a consistent way. All interviews and group discussions were conducted in Kurdish or Arabic, audio recorded and transcribed and translated. Observation notes were taken by hand and entered into a digital spreadsheet the same day. The handwashing demonstrations were video recorded.

Preliminary data analysis was done concurrently with data collection. This allowed us make theoretical and methodological notes [30] and decide when we had reached a point of

**Table 2. Summary of the number of participants per method.**

|  | All sites | Nargazliya Camp | Sheikhan Camp | Villages |
|---|---|---|---|---|
| **Unstructured Observation** | 20 households | 7 households | 7 households | 6 households |
| **Focus Group Discussions** | 10 groups | 6 groups* | 2 groups | 2 groups |
| **In-depth Interviews** | 98 participants | 26 participants | 33 participants | 39 participants |
| **Handwashing Demonstrations** | 24 participants | 6 participants | 6 participants | 12 participants |

Note: Some people participated in more than one method.

*Group discussions had to be smaller in size in this camp.

saturation. Audio recordings from interviews and group discussions were transcribed and translated. Methods with ranked or scaled data were summarised in spreadsheets. Drawings, photos and videos were descriptively summarised. All data was imported into NVivo 12 software. The data analysis was informed by the process outlined by Braun and Clarke [31]. Data were classified according to study site, gender and method and comparisons were made between study sites and between current practices and reported behaviours prior to the conflict. A top-down coding framework was applied based on the BCD checklist and emergent themes were added. The coding was primarily done by SW, with sub-set of coding cross-checked for validity by TH, DI and WKI.

## Ethics and consent

Informed written consent was obtained from each participant. Ethics approvals were given by the London School of Hygiene and Tropical Medicine (Protocol 13545) and Hawler Medical University. Permission was also granted by the Board of Relief and Humanitarian Affairs in Kurdistan. Organisations working in the area were informed of our work and preliminary findings were shared immediately after data collection to enable utilisation within programmatic work.

## Results

In total, 159 people took part in this research—58 in Nargizlia Camp, 49 in Sheikhan Camp and 52 across the two villages. Most respondents were women (68%) reflecting both that domestic hygiene often fell to the female household head and that many households did not have a male household head due to the conflict. Patterns of displacement reflected the complexity of the crisis, with many people being displaced multiple times. Table 3 summarises the characteristics of the sample.

WASH access was in line with the Sphere Humanitarian Standards [32]. Most families in the camps had received hygiene kits at some point (100% in Nargizlia and 98% in Sheikhan) and had been exposed to hygiene promotion (93% in Nargizlia and 78% in Sheikhan). In contrast, 32% of participants from the villages had never received kits and 61% had not been exposed to hygiene promotion. S4 Appendix provides greater detail on exposure to hygiene kits and promotion.

## Behaviour

While this research intended to focus on handwashing behaviour, participants commonly conceptualised handwashing as linked to a broader set of household cleaning behaviours rather than as a stand-alone behaviour.

When asked, all participants reported they would always wash their hands with soap after the toilet or before preparing or eating food. Household observations confirmed hand cleaning was frequent aspect of daily routines in the camp settings but varied by household within village settings. However, at critical handwashing times, handwashing with soap was relatively low across all three settings (27–29%). S5 Appendix summarises observed handwashing behaviours across 20 households participating in this method.

## Handwashing determinants unaffected by displacement

We identified 33 specific determinants influencing handwashing behaviour in these contexts. Of these, 12 appeared to be unaffected by displacement (i.e. these determinants had a similar influence on behaviour pre and post the conflict). Social approval, the motives of status and

**Table 3. Socio-demographic characteristics of all participants across the two camps and two villages.**

| Socio-Demographic characteristics | Total N = 159 | Nargazliya Camp N = 58 | Sheikhan Camp N = 49 | Villages N = 52 |
|---|---|---|---|---|
| **Gender** | | | | |
| Female | 108 (68%) | 41 (71% | 31 (63%) | 36 (69%) |
| **Religion** | | | | |
| Muslim | 112 (70%) | 58 (100%) | 0 (0%) | 52 (100%) |
| Yazidi | 47 (30%) | 0 (0%) | 49 (100%) | 0 (0%) |
| **Ethnicity** | | | | |
| Arab | 92 (58%) | 55 (95%) | 0 (0%) | 37 (71% |
| Kurdish | 50 (31%) | 1 (2%) | 49 (100%) | 0 (0%) |
| Turkmen | 2 (1%) | 2 (3%) | 0 (0%) | 0 (0%) |
| Shabak | 15 (9%) | 0 (0%) | 0 (0%) | 15 (29%) |
| **Literacy** | | | | |
| Literate | 95 (60%) | 37 (64%) | 29 (59%) | 29 (56%) |
| **Household Size** | | | | |
| Average | 8.7 | 6.8 | 8.2 | 9.62 |
| Range | 2–28 | 2–18 | 2–16 | 3–28 |
| **Displacement status** | | | | |
| Internally Displaced | 113 (72% | 58 (100%) | 49 (100%) | 8 (15%) |
| Returnee | 43 (27%) | 0 (0%) | 0 (0%) | 42 (81%) |
| Host community (did not leave) | 2 (1%) | 0 (0%) | 0 (0%) | 2 (4%) |
| Period since displacement (range) | 2 weeks—3 years | 2 weeks– 6 months | 6 months– 3 years | 2 weeks– 5 months |
| Period since return (range) | 1 day—1 year | - | - | 1 day—1 year |
| **How many people share the same toilet** | | | | |
| Average | 10.3 | 11.3 | 8.5 | 11.6 |
| Range | 2 to 28 | 5–18 | 2–16 | 3–28 |

disgust, and being female reportedly had a positive influence on behaviour prior to displacement and appeared to continue to be influential across all sites after the conflict. Prior to and post displacement, children were thought to need parental support to practice handwashing and older people were perceived to face barriers to handwashing because of reduced mobility. several determinants outlined within the BCD framework did not appear to facilitate or deter handwashing behaviour before or after displacement. These included determinants related to the biological environment (e.g. the presence of insects and snakes), literacy or education levels, employment status, ethnicity and religion, the motive of fear and knowledge about disease transmission. Across all sites, handwashing knowledge was high, with 99% of participants being able to explain disease transmission and believing that handwashing had health benefits. Participants perceived personality to be one of the strongest determinants of handwashing. Multiple participants explained that if you were hygienic before the crisis, then you would continue to be hygienic when displaced.

### Handwashing determinants affected by displacement

The remaining 21 determinants appeared to have a different influence over handwashing behaviour in the displacement period. In the sections below we describe these patterns against the determinant categories of the BCD framework, bringing together findings from across the research methods.

**Characteristics and capabilities.** In general, the influence of personal characteristics on handwashing was unaffected by the conflict, however larger families were less able to practice

handwashing in the post displacement period. In the camps, larger families reported that hygiene kit products ran out quickly. In the villages multiple families were often sharing one house because of the destruction and this made it harder to maintain hygiene.

Participants felt handwashing was easy and within their capabilities. However, without prompting 25 participants said they had been experiencing mental health challenges because of the conflict. For some people this meant they felt less able to undertake daily tasks, including those related to hygiene:

*"We have difficulties with psychological problems, otherwise if we didn't have this challenge, we could be more clean within our homes, and in the way we look and everything."*–Woman living in Nargizlia Camp

Others explained handwashing had become a coping mechanism that made them feel more at ease and aided them to manage their trauma and worries:

*"I feel comfortable when I wash my hands. . .If I don't wash them I feel like it affects me and I start feeling more worried and stressed"*–Woman living in a village

**Physical and social environment.** The physical environment includes factors in the natural or built environment, climate and geography. The physical environment had a substantial influence on behaviour because it was so different to the circumstances people were accustomed to prior to the conflict, however this was more pronounced in the camp settings. In the camps there were three interlinking aspects of the physical environment which heightened the frequency of handwashing behaviour. These were the perceived dirtiness of the camps, the tented shelters which were hard to keep clean, and the extreme summertime heat.

In the camps people described their settings as dirty, with some people expressing that their living circumstances were so disgusting that they felt like animals:

*"Here is no place to stay as a human. As much as we can, we clean, but it is still dirty, there is not enough soap. . .our bodies are not clean and not comfortable. . . it has affected me a lot and now I feel we are animals, not human."*–Woman living in Nargizlia

Camp residents explained it was not easy to adapt to living in a tent and it required them to spend much of the day cleaning:

*"Before we were living in nice houses, none of us have ever lived inside a tent before so it's a big change in the environment. Before it was easy for us to clean the ceramic tiles in our house but now our floor is made from dust and our walls from plastic."*—Man living in Nargizlia Camp

The weather was mentioned as a challenge by all participants in the camps. Residents were used to the summer heat but felt less able to cope with it in the camp settings and were observed to regularly splash their hands, face, feet and bodies with water to cool down. This new behaviour appeared to deter handwashing with soap at critical times as hands were perceived to have been cleaned recently.

In the villages people did not perceive their physical environment to influence handwashing. Participants felt relieved to be back home and reported that their lives and their behaviours had gone back to normal:

*"We were happy when we returned. . . now we feel safe again and even though many things have changed, like this room [she points to a crack running the length of the wall] we are able to do all of the behaviours we were used to doing"*–Woman living in the villages

Some participants did report they had to clean more frequently because houses and water systems were damaged during the conflict. However, observations in the villages indicated cleaning and handwashing was done less frequently than in the camps.

The social environment includes people's social networks, and how they perceive themselves within these. It also covers how people socialise and influence with others. One of the participatory methods involved mapping social networks before and after the conflict. This indicated that people's social networks decreased in size with displacement and that key relationships, such as close friends and extended family, were lost. Both these groups were reportedly important for supporting good behaviours prior to the conflict. People in the camp settings also reported choosing to be less social. Despite this lack of sociality, the densely populated living environments in both camps meant people did notice the hygiene behaviour of others. However, the lack of personal connection and the recognition that people had been through difficult circumstances meant people would avoid reminding others to be more hygienic:

*"No one would say anything [about whether I wash my hands or not] because they don't know me. If I don't know them, what could I say, I can't correct their behaviour either."*–Woman living in Nargizlia

*"In [my home town] it was not a big deal to remind your family or friends to be hygienic . . . and they will pay attention and follow you. But here if people do the wrong thing, then I would be afraid to tell them, it's difficult here, I would be afraid they would do something like suicide."*–Man living in Shikhan Camp

In contrast, the built environment of the villages meant handwashing was not able to be noticed between neighbours:

*"Everyone is in his house when he is doing those things [handwashing] so no one knows what you are doing."*–Woman living in the Villages

**Behavioural settings.** Behavioural settings incorporate the proximal aspects of the social environment (roles, norms and routines) and physical environment (the 'stage', props and infrastructure) that result in regular sequences of behaviour, and which enable or prevent handwashing from taking place in the settings where it needs to happen (e.g. kitchens or toilets) [29].

The characteristics of the physical locations where handwashing took place were different across the three settings, but relatively homogenous within each setting. In Nargizlia Camp people washed their hands in either the kitchen or the bathroom. Taps had been established for purposes other than handwashing and this meant people had to bend over when trying to wash hands. Given that WASH facilities were shared, families often kept soap away from these facilities, inside their tents. Households only had one type of soap (distributed by NGOs) and this was used for all purposes. In some cases, shared WASH facilities acted as a barrier to handwashing:

*"When my wife is going to wash her hands in the kitchen she is always thinking she should hurry up because her neighbour is waiting their turn."*–Man living in Nargizlia Camp

In Sheikhan Camp hands were washed in settings similar to Nargizlia Camp. However, in Sheikhan Camp facilities were not shared and consequently most families had personalised these spaces so that handwashing was easier to practice. This included adding mirrors, soap dishes and seats to enable handwashing in spaces where taps were positioned at a low height. In Sheikhan there was a more diverse array of soaps available which were used for different purposes:

> *"We buy different types of soap, for laundry we use the powder, we buy liquid detergent for dishes, for showering we buy the shampoo and soap with a nice smell and for handwashing we buy this bar soap."*—Man in Sheikhan Camp

In villages people were accustomed to washing their hands at porcelain basins with piped water. These facilities often had mirrors placed above them, liquid soap dispensers or soap dishes. Such facilities were located outside bathrooms or at entrances to houses. Only one of these facilities was observed to be in working order at the time of the research as most were damaged during the conflict or were no longer connected to piped water. Instead, most families now washed their hands by pouring water from a jug. In villages soap was scarce, with some households not having any soap and others using laundry powder or shampoo for handwashing.

Participants involved in group discussions across all research sites expressed similar desires in relation to handwashing infrastructure. People felt that mirrors above the facility, liquid soap and a basin to catch wastewater were the design factors most likely to increase handwashing frequency. In Nargizlia having private facilities also emerged as a priority. A full summary of these results is provided in S6 Appendix. Participants explained that since displacement, the primary factors influencing their decision-making around soap were cost, availability and how well the soap lathered.

In all three settings cleaning took up a greater proportion of day-to-day routines since displacement. Largely, this was because tents and damaged buildings were hard to keep clean, however in the camp settings cleaning also took place due to a lack of alternative pastimes:

> *"Back home it didn't feel like we had a set routine, every day we had different duties. But here in the camp every day is the same routine—breakfast then wash dishes and clean, lunch time then wash dishes and clean, then dinner, it's just the same thing in repetition."*–Woman in Sheikhan Camp

The frequency of cleaning-related activities throughout the day was observed to be associated with a decreased likelihood of handwashing with soap at critical times. This was because hands were often washed in conjunction with these other cleaning tasks instead.

A person's roles, identity or perceived responsibilities can shape their handwashing practice and the extent that they encourage this behaviour among others. Participants reported that they felt they had acquired a new 'label' of being an IDP and this was associated with a perceived loss of agency and sense of individuality:

> *"Everything was in our control before, nothing seemed difficult but when you become an IDP it's not within your ability to control the situation. You have to start from zero. When we were displaced, I had to ask NGOs even for the most simple things. Before I would never dream an organisation would have to provide me with rice or soap."*–Man living in the villages (describing a period when he was displaced to a camp)

This 'IDP label' made people in the camp settings feel like they were less able to practice handwashing and less able to support their children to be hygienic. In contrast one man living in the villages explained because he was an IDP, others in the village would not see his family as being 'like them' and would assume his family were unhygienic. Consequently, he felt he had to remind his children to always wash their hands and look nice so they would be accepted in the community.

One of the participatory activities in the interviews was designed to understand social norms and social expectations around handwashing. Handwashing was seen to be a socially desirable norm across all settings, with all participants saying that if you asked 100 people within their area whether they wash their hands with soap at critical times, they would all say yes. Accordingly, participants also felt people would judge them negatively if they did not wash their hands. However, in all three sites, participants questioned whether handwashing was a descriptive norm, meaning that participants felt it was not always performed by others. People in the camp settings mentioned handwashing behaviours were influenced by neighbours mimicking each other's behaviour as they tried to fit in:

> *"Some people care about hygiene and some not so much. If someone is not hygienic and you visit them, then this will affect you too, because people here mimic their neighbour's behaviour more than back home."*–Woman living in Sheikhan Camp

**Cognitive determinants.**   When participants were asked about the hygiene challenges they experienced since displacement, no one spontaneously mentioned handwashing. When we encouraged people to rank handwashing in relation to their other hygiene challenges, it was ranked last by most of the participants. Despite this handwashing was valued by participants:

> *"I am always washing my hands with water and soap. In our family the most important thing is hygiene."*—Man living in Sheikhan Camp

Participants were observed to make trade-offs in relation to the costs, benefits and ease of practicing handwashing with soap. The most influential determinant in this category was the cost of having soap in sufficient quantities as people had experienced changes to their income due to the conflict. In Nargizlia Camp an informal system emerged to allow people to exchange other items for soap:

> *"When hygiene kits are delayed and I can't buy these things, our children will be dirty and not clean and they will get diseased. . .the only thing we can do is to sell our food and buy these hygiene items but we are not supposed to do that."*–Man living in Nargizlia Camp

In Sheikhan Camp the cessation of hygiene kit distributions was a source of worry. The population used most of their income to buy hygiene products:

> *"We have a problem in that money is not enough because we only earn 50,000 Iraqi Dinar and easily all the money can get spent on soap and detergent."*–Man in Sheikhan Camp

In the villages, markets had not resumed and so people had to travel further and pay more for soap. Across all sites these circumstances led to people conserving soap or buying poorer quality soap than they would have done prior to the conflict, making handwashing less desirable to practice:

*"We used nicer quality soap before Da'ish and we remember that nice experience, but now we can't buy them [nice soap] because we don't have enough money."*–Woman living in the villages

*"You know it doesn't mean we are not clean, it's just that sometimes we don't have hygiene kits so then we have to use soap less and preserve some in order to control our lives."*–Woman living in Nargizlia Camp

In the villages, people were observed to just rinse their hands without using soap, while others described skipping showers to conserve soap. In the camps soap was prioritised ahead of other needs (e.g. food) and people stockpiled of soap.

Illnesses associated with handwashing–such as diarrhoea–were of limited concern to respondents across all three sites. Participants were concerned with chronic health conditions, skin diseases and mental health.

Diarrhoea-related risk perception varied across the study sites with participants in Nargizlia Camp perceiving that their children were more likely to get diarrhoea now they are living within the camp as compared to prior to displacement. Participants in Nargizlia Camp were also more likely think diarrhoea was a major cause of concern and that it could have a serious impact on the whole family. In Sheikhan Camp, participants perceived their risk to be greater than prior to displacement but acknowledged people within the camp were generally hygienic so this minimised the risk. Participants in the villages were the least concerned about diarrhoea and felt that even if their children got diarrhoea, it was unlikely to cause serious illness or death. People in the villages did not perceive that their likelihood of getting diarrhoea had increased in comparison to prior to the conflict. There was agreement across the research sites that diarrhoea could sometimes be prevented through handwashing, and in the two camp settings people did report increasing their handwashing frequency because of their perceived increased risk. A heat map of the responses to risk-related questions is available in S7 Appendix.

Participants were asked to describe motive-based responses to handwashing scenarios. Motivational responses across all study sites were relatively similar. There was consensus that handwashing at critical times was associated with being a respectable person (status), disgust and comfort. Nurture ('that person would be a good parent'), affiliation ('that is the kind of person I would want to be friends with'), attract ('I would find that person attractive') and fear were less associated with handwashing.

In interviews, discussions and observations in the camp settings hygiene behaviour was triggered by a desire to feel comfortable and 'fresh' despite their surroundings. During observations in the villages, hands were typically only washed with soap when they were visibly dirty or smelly.

**Handwashing determinants before and during displacement.** Overall, we identified 33 specific determinants through our research and grouped these within the 16 'determinant categories' outlined by the BCD framework. Table 4 synthesizes findings from all data collection activities and summarises the reported and observed determinants of handwashing behaviour prior to displacement and across the three research sites.

## Discussion

This research revealed that the determinants of hygiene behaviour do appear to differ in the wake of a conflict and that the influence of certain determinants is contingent on the type of setting people are displaced to. Populations in this study were aware of the health benefits of hygiene, and handwashing was found to be valued and a socially approved norm. However,

**Table 4. Associational relationships between identified determinants and handwashing with soap at critical times.**

| Determinants | | Prior to displacement | Nargizlia Camp (Early displacement) | Sheikhan (Longer-term displacement) | Villages (Returnees & IDPs) |
|---|---|---|---|---|---|
| **Behavioural measure** | | Self-report | Self-report & observation | Self-report & observation | Self-report & observation |
| **Individual characteristics** | Gender (being female) | + | + | + | + |
| | Being a child or an older person who needs support to wash hands | - | - | - | - |
| | Ethnicity | o | o | o | o |
| | Religion | o | o | o | o |
| | Personality type–being a neat person | + | + | + | + |
| | Literacy | o | o | o | o |
| | Large family size | o | - | - | - |
| **Capabilities** | Having mental health challenges | o | -\+ | -\+ | -\+ |
| **Physical environment** | Hot weather | o | + | + | o |
| | Living environments are perceived to be dirty and hard to clean | o | + | + | o |
| **Social Environment** | Social support to encourage handwashing | + | - | - | + |
| | Sociality and interaction with others | + | - | - | o |
| | Social judgement or social sanctions if handwashing is not practiced | + | o | o | o |
| **Biological Environment** | Presence of insects or vermin | o | o | o | o |
| **Stage** | Using shared WASH facilities | o | - | + | - |
| **Infrastructure** | Having sufficient access to water availability | + | + | + | - |
| | Water quality (hard or salty water which prevented lathering) | o | o | - | - |
| | Having a dedicated handwashing facility | + | + | + | -/+ |
| | Having a soap dish or dispenser | + | - | + | - |
| **Props** | Having access to sufficient quantity of soap | + | - | + | - |
| | Having access to sufficient quality of soap | + | o | + | - |
| **Roles** | Being an IDP | + | - | - | - |
| **Routine** | Frequency of water-related cleaning activities* | o | - | - | o |
| | Having other pastimes | - | + | + | o |
| **Norms** | Handwashing is seen as something that is socially approved | + | + | + | + |
| | Perceived handwashing practices of neighbours, friends and family | + | + | + | o |
| **Executive Brain** | Knowledge about the health benefits of handwashing | o | o | o | o |
| | Perceived severity of diarrhoeal disease | - | + | + | - |
| | Perceived vulnerability to diarrhoea disease | o | + | + | o |
| **Discounts** | Cost of soap | + | + | o | - |
| **Motivated Brain** | Fear | o | o | o | o |
| | Nurture | + | o | o | o |
| | Status | + | + | + | + |
| | Comfort | o | + | + | o |
| | Hoard | o | - | - | - |
| | Disgust | + | + | + | + |

Determinants which positively influence behaviour are coloured green (+), those which have no effect on behaviour are allocated coloured grey (o) and those with a negative effect on behaviour are coloured red (-). Where results were mixed or varied at an individual level they are yellow (-/+).

* This negatively effects handwashing at critical times but increases hand rinsing and handwashing with soap at non-critical times.

among the competing priorities of populations in this study, handwashing was not defined as a problem in the way that other social, economic or health challenges were.

We observed that different behavioural patterns were associated with different physical settings that populations were displaced to. In the camp settings a new 'hyper-hygienic norm' formed, driven by a heightened perceived risk of disease and a desire to create order, comfort and cleanliness within challenging living environments. The hyper-hygienic norm led to an increased amount of time being spent on cleaning-related activities and an increased interaction with soap and water, but it did not always result in hands being washed at critical times. Participants who had returned home to their villages post-displacement, felt relief at being somewhere familiar and 'safe' and this prevented participants from seeing that physical damage to infrastructure (caused by the conflict) created new disease risks. Positive memories of 'home' are thought to be a strong motivator for displaced persons to return home following conflict [33,34], and may in the short-term blind people to the extent of change that has occurred in these locations, affecting people's likelihood of prioritising protective behaviours like handwashing.

In addition to the physical settings, we found that many of the determinants which detrimentally affected hygiene behaviours, relate to the broader psychological, social and economic consequences of conflict. Mental health was voluntarily reported in our study, but our findings indicate that depression and trauma associated with conflict and displacement may cause some people to deprioritise handwashing, while others may increase their handwashing to cope with their circumstances. Research in stable settings has found students with depression had lower rates of handwashing [35,36] and that experiences of disgust, discomfort, trauma and stress can cause the emergence, or worsening, of obsessive-compulsive handwashing [37–40]. The association between mental health and hygiene warrants further research in humanitarian crises.

Our research also found that, during displacement, hygiene behaviour is influenced by disruptions to people's sense of identity and their perceived agency. Participants had a heightened awareness about what humanitarians and others may assume about their hygiene behaviour because they were labelled as 'IDPs'. Participants used handwashing and other observable hygiene behaviours (e.g. tidiness of households) as a way to assert some degree of control over the unpredictability of their circumstances and their new physical environment. Anthropological and geographical work in conflict-affected settings has described how displaced persons regain agency and subvert imposed identities by creating order and homeliness in otherwise hostile environments and by maintaining routine, ordinary behaviours from their place origin [41–46]. Handwashing appears to be part of this set of 'ordinary restorative behaviours'. In our study we found handwashing also became a 'social indicator'–a visible way of demonstrating good values and enabling people to gain acceptance and fit in to their new social environments. This makes sense given that displaced populations are often faced with a dissolution of pre-existing social orders and norms and have to navigate new relationships among unfamiliar neighbours [47,48].

Our findings align with existing literature about handwashing during crises. For example, research found that the availability of soap, water and handwashing facilities were a key determinant of handwashing practice. Actual or perceived scarcities of water and soap and the deprioritisation of these items for handwashing (as opposed to other household tasks) have been identified as common challenges in the wake of humanitarian crises [49–52]. The availability of desirable and conveniently located handwashing facilities is thought to be one of the most influential determinants of hygiene behaviour in stable settings [7,12,53] and improving handwashing infrastructure may have an even greater influence among displaced populations [51,54,55].

Our study found handwashing was most strongly influenced by motives of disgust, status and comfort. Other studies among displaced populations have suggested motives of nurture and affiliation should be utilised by hygiene programmes in humanitarian settings [51,56,57]. The use of emotional drivers in hygiene programming remains contentious. If motives are used with a lack of contextual understanding, programmes could lead to stereotyping and stigmatisation [58–60]. Our results indicate motives like nurture and affiliation should be used with caution given that participants felt less able to care for their children during displacement and felt disconnected from the social group to which they belonged. Variation between our results and other studies among displaced populations could also be due to inconsistent methods for assessing motives. For example, most studies focus on the general appeal of one motive over another [51,56], while our study specifically explored motives in relation to the target behaviour. More work is needed to assess the validity and reliability of methods for assessing motives.

## Recommendations

As in stable settings, programmes which focus only on imparting hygiene information are likely to be insufficient to create change in conflict-affected settings where knowledge is already high and there are numerous competing priorities [7,12]. Improving handwashing in the camp settings could draw attention to new norms and link these to critical handwashing occasions. There are opportunities for hygiene promotion programmes to contribute to re-building a sense of individuality and agency among IDPs. For example, in Sheikhan Camp some flexibility in WASH services had allowed participants to personalise and decorate their WASH facilities–actions which encouraged and enabled behaviour. Programming could easily incentivise the customisation of handwashing facilities so to position handwashing as a behaviour that is desirable, pleasurable and refreshing. In the out-of-camp settings hygiene programmes could draw attention to new risks in the physical environment and heighten disgust in relation to these, as both can be powerful drivers of hygiene behaviour [53,61–63]. Programming could provide social and financial incentives to encourage families to re-build damaged handwashing facilities and thus cue behaviour [7,12]. Most importantly, WASH actors should not view hygiene as a narrow public health issue but rather design behavioural assessments which explore a range of determinants, including aspects of the physical and social environment. Hygiene programme design could be strengthened by integration with psychosocial support and livelihoods initiatives.

## The utility of qualitative research in crises

The use of multiple rapid qualitative methods was feasible in these displacement settings because behaviour was relatively homogenous within each study site, allowing us to reach saturation quickly. We conducted these methods alongside a more traditional survey-based approach to assess behaviour determinants [21]. While the results between the two approaches showed some consistency, the qualitative approach generated data which is likely to be more useful for informing programme design.

## Limitations

There is no agreed way of understanding and measuring the determinants of handwashing behaviour [12]. In our study we relied quite heavily on self-reported perceptions of the determinants that influenced current behaviour and recall of the factors that influenced behaviour in the past. Given that handwashing is a socially desirable behaviour these self-reported perspectives are likely to be biased with people presenting more favourable versions of their

behaviour [27,64]. 'Talk-based' methods such as interviews and group discussions are also likely to generate a partial view of behaviour given that people are typically less able to describe sub-conscious determinants of behaviour [29,65]. We also asked participants to recall their handwashing behaviours prior to the crisis and long-term recall like this is known to be prone to errors and misremembering [66,67]. To compensate for these limitations, we triangulated data across the methods used and employed visuals, props and scenarios as part of participatory activities in order to illicit different types of responses. We also complemented talk-based approaches with observations and demonstrations. Some of the activities we used within interviews and group discussions, were also developed for this research or have only been piloted in a small number of other studies. Therefore, the validity and reliability of these tools should be tested further. Given the methodological limitations we were working with, our research was only able to describe apparent associational relationships between determinants and handwashing behaviour, rather than quantify or definitively state the impact of determinants on behaviour.

This research was conducted in partnership with Action Against Hunger (AAH) and our research team were required to wear a branded vest throughout data collection. Given that AHH have a history of working on WASH projects in this region, respondents may have given socially desirable answers or behaved differently during observations given their contextual awareness of AAH's role. In our daily reflection sessions, we actively discussed our individual and collective positionalities and how this may have shaped the research.

While our research sought to be purposively select research sites to consider different types of post-conflict displacement, contextual factors may limit the generalisability of these findings. Therefore, additional research exploring the determinants of handwashing behaviour in other displacement settings would be merited.

## Conclusion

Our findings strengthen the evidence base on handwashing determinants in the post-conflict displacement period. Our work supports prior research in that it suggests programmes are likely to be most effective if they go beyond hygiene education and instead try to overcome a range of behavioural barriers. Variations in the physical environment and WASH services within each of our research sites also point towards opportunities for humanitarians to shape behaviour by creating enabling infrastructure and providing access to desirable products. Lastly our work highlights the importance of treating behaviour holistically and integrating hygiene programming into other sectors.

## Supporting information

**S1 Appendix. Handwashing determinant definitions adapted from on the BCD checklist of determinants and accompanied by method selection.**
(DOCX)

**S2 Appendix. Purpose, description and sample size for each of the methods done within group discussions.**
(DOCX)

**S3 Appendix. Description and sample size for all methods done at a household or individual level.**
(DOCX)

**S4 Appendix. Exposure to hygiene promotion among interview participants.**
(DOCX)

**S5 Appendix. Summary of household observations of handwashing.**
(DOCX)

**S6 Appendix. Handwashing facility design factors that group discussion participants thought would be most likely to increase their handwashing behaviour.**
(DOCX)

**S7 Appendix. Heat map of scaled group discussion responses to diarrhoeal risk related questions.**
(DOCX)

## Acknowledgments

We would like to thank the following people for helping to facilitate the research and contributing to ongoing reflections about emergent insights: Waleed Rasheed, Honar Hasan, Asmaa Farooq, Basima Ahmed, Aso Zangana, Mustafa Abdalla, Nazar Shabila, Tara Vernon, Geraldine Delestienne, Karine Le Roch and Jean Lapegue. We would particularly like to thank the individuals who gave of their time to participate in this research and who welcomed us into their homes and shared their personal experiences so openly.

This research was undertaken as part of the Wash'Em Project which aims to improve handwashing promotion in humanitarian crises. The research was made possible by the generous support of the American people through the United States Agency for international development's Bureau of Humanitarian Assistance. The contents are the responsibility of the authors of the paper and do not necessarily reflect the views of USAID or the United States Government.

## Author Contributions

**Conceptualization:** Sian White.

**Data curation:** Sian White.

**Formal analysis:** Sian White.

**Funding acquisition:** Sian White, Thomas Heath.

**Investigation:** Sian White, Waleed Khalid Ibrahim, Dilveen Ihsan.

**Methodology:** Sian White, Val Curtis, Robert Dreibelbis.

**Project administration:** Sian White, Thomas Heath.

**Supervision:** Thomas Heath, Karl Blanchet, Val Curtis, Robert Dreibelbis.

**Validation:** Thomas Heath, Waleed Khalid Ibrahim, Dilveen Ihsan.

**Visualization:** Sian White.

**Writing – original draft:** Sian White.

**Writing – review & editing:** Thomas Heath, Waleed Khalid Ibrahim, Dilveen Ihsan, Karl Blanchet, Robert Dreibelbis.

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
