## [Decision Letter · Decision Letter 0]

9 Dec 2021

PONE-D-21-15601How does hygiene behaviour change over the course of displacement? A qualitative case study in Iraq and Kurdistan.PLOS ONE

Dear Dr. White,

Thank you for submitting your manuscript to PLOS ONE. After careful consideration, we feel that it has merit but does not fully meet PLOS ONE’s publication criteria as it currently stands. Therefore, we invite you to submit a revised version of the manuscript that addresses the points raised during the review process. Please consider constructive comments from both reviewers (feedback from one is detailed below, the other as an attachment). Please submit your revised manuscript by Jan 23 2022 11:59PM. If you will need more time than this to complete your revisions, please reply to this message or contact the journal office at plosone@plos.org. Please include the following items when submitting your revised manuscript:A rebuttal letter that responds to each point raised by the academic editor and reviewer(s). You should upload this letter as a separate file labeled 'Response to Reviewers'.A marked-up copy of your manuscript that highlights changes made to the original version. You should upload this as a separate file labeled 'Revised Manuscript with Track Changes'.An unmarked version of your revised paper without tracked changes. You should upload this as a separate file labeled 'Manuscript'.

We look forward to receiving your revised manuscript.

Kind regards,

Hannah Tappis, DrPH, MPH

Academic Editor

PLOS ONE

Journal Requirements:

5.  Please amend your list of authors on the manuscript to ensure that each author is linked to an affiliation. Authors’ affiliations should reflect the institution where the work was done (if authors moved subsequently, you can also list the new affiliation stating “current affiliation:….” as necessary).

Reviewers' comments:

Reviewer's Responses to Questions

**Comments to the Author**

1. Is the manuscript technically sound, and do the data support the conclusions?

Reviewer #1: Yes

Reviewer #2: Partly

2. Has the statistical analysis been performed appropriately and rigorously? 

Reviewer #1: N/A

Reviewer #2: N/A

3. Have the authors made all data underlying the findings in their manuscript fully available?

Reviewer #1: Yes

Reviewer #2: No

4. Is the manuscript presented in an intelligible fashion and written in standard English?

Reviewer #1: Yes

Reviewer #2: Yes

5. Review Comments to the Author

Reviewer #1: I would like to congratulate the authors for this good piece of work that is very important in the field of hand hygiene. The findings will add on the existing knowledge and may be applied across the globe in the displaced society. With the current state of the devastating Covid-19, its worthy to consider publishing the article in Plos One so as to be accessed by the wider community. Hand hygiene is one of the intervention much touted for Covid-19 prevention. However, there are a some minor comments that you may need to work on. See details in the attachment.

Reviewer #2: Summary: Handwashing with soap at key times is a key measure to reduce infectious diseases, such as diarrhoeal and respiratory infections. Although displaced people are particularly vulnerable to these diseases, knowledge about the determinants of handwashing in such a setting is limited. The manuscript contributes to closing this major research gap and thus to informed and evidence-based programming in such settings. As such, the manuscript makes an important contribution to research and practice and should be published, pending on some major revisions. These include a clearer definition of the manuscripts scope, a more precise description of the methodology and of the applied comparisons, and a re-structured presentation of the findings, which presents these comparisons more clearly and helps the reader more to digest the information. These points are explained in more detail below. When revising, the authors should check spelling and punctuation – the amount of typos is on the edge to unacceptable.

General comments:

• The scope of the study could be better defined, namely in title, abstract and throughout manuscript:

o Displacement, post-conflict settings, crisis-affected settings are all mentioned but may imply different things. I suggest to explain more precisely whether you researched and/or have findings relevant for crises more generally or only when people are displaced, any form of displacement or only when displaced due to conflicts.

o The phrasing of “change over the course of displacement” suggests a longitudinal design but it seems the design was not longitudinal. I suggest to phrase more precisely to avoid readers having inadequate expectations. Also, it is not entirely clear how it co-occurs with the different settings or whether there is both the time-component and the setting component

• Research on habit formation suggests that disruptive events may help change behaviour. It might be worthwhile reflecting on what this may mean for the manuscripts context. Check out research by Bas Verplanken.

• Methodology should be extended. See some specific comments on issues particularly regarding collection of “prior” data.

• The results section is quite dense and it is not entirely clear what the comparison criteria were. See specific comments on the latter point. I encourage the authors to reflect on whether the results could be reported in a different way that eases the processing of the information and makes the comparisons clearer.

• Quite a lot of typos, including punctuation. Also some inconsistencies in phrasing or use of terms. Please check spelling and punctuation when revising.

Specific comments:

Abstract: not really clear what is meant with type of setting and whether “three post-conflict settings” are these different types or again another setting. Maybe rephrase and explain a bit more what was researched where.

Line 52: maybe inform which determinants were under reported.

Lines 59 and 64 might profit from some additional references on determinants in crisis-settings (first two) and over-estimation (last one)

Contzen, N., & Mosler, H. J. (2013). Impact of different promotional channels on handwashing behaviour in an emergency context: Haiti post-earthquake public health promotions and cholera response. Journal of Public Health (Berlin, Heidelberg), 21(6), 559-573. doi:10.1007/s10389-013-0577-4

Contzen, N., & Mosler, H. J. (2015). Identifying the psychological determinants of handwashing: results from two cross-sectional questionnaire studies in Haiti and Ethiopia. American Journal of Infection Control, 43(8), 826-832.

Contzen, N., De Pasquale, S., & Mosler, H. J. (2015). Over-reporting in handwashing self-reports: potential explanatory factors and alternative measurements. PLoS One, 10(8), e0136445 (22 pp.). doi:10.1371/journal.pone.0136445

Line 88: IDPs is used without term being introduced.

Line 87-90: not clear whether IDPs are also Arab or Shabak or something else, whether they are part of the “residents” displaced and returned, or whether they were additional, whether they were housed in other people’s houses or had their own which were also damaged etc. Explain more precisely.

Line 97: what type of characteristic is meant here??

Line 106: any action or only actions relevant to handwashing?

Line 112 and 117: not clear what the participatory activities were. Even if there is an SM explaining in detail, maybe say 1-2 sentences in the text to give readers a basic understanding on what had been done.

Methods section: IDIs and FGDs are really hard to read. Maybe use interviews as abbreviation of IDIs and group discussions as abbreviation of FGDs.

Table 3: for all dichotomous variables half of the information is redundant

Results section: determinants are said to “influence” behaviour or have an “impact”. To make any statements about influence or impact, at least a longitudinal design or better an experimental design is needed. Based on the data available, statements such as “seem to support or hinder” or “are related with” are possible, but nothing more.

Line 176: you refer to 12 determinants but in the rest of the paragraph it is not that clear what is a determinant and what is “other information” you provide along. Restructure and report more precisely.

Line 179, 181-182: “prior to displacement“, “before nor after displacement“ suggests you asked respondents how it had been before and how it is now. This was not clear to me based on method section and raises some questions as collection of retrospective data has quite some limitations. Please explain in more detail in methods section.

Results on determinants unaffected and affected by displacement: it is not entirely clear to me whether the authors conclude on whether or not a determinant is unaffected by displacement based on respondents statements about “before” and “after” (see above comment) or based on differences between settings. This is in line with a previous comment that it is not entirely clear whether both a time- and a setting-component were researched and how they are disentangled. The same problem occurs again in section on determinants before and during displacement � based on title not clear whether this is now another analysis or a summary. I strongly suggest the authors revise and describe more precisely. Including being more precise on what your comparison criteria are.

Line 354: injunctive norm is introduced but then the example which is given (if you asked 100 people within their area whether they wash their hands with soap at critical times) is rather an example of either descriptive norms or maybe of socially desirable responding. Check what concept you really mean here and whether the example fits to it or not.

Line 358: being afraid that one would be judged is an aspect of injunctive norm � see previous comment � section on norms seems to need some revision.

Line 367 and following: whether something is a challenge or not does not say anything about whether it is important or not. Something can be a challenge but not important to me, or not a challenge but very important to me � not clear what this paragraph really taps into and what the key message is and how it is based on any of the statement by participants. Same issue again in line 453 and following.

Table 4: some information not very clear or useful. E.g., Age (being a child or an older person) has a negative “effect”. What does that mean exactly? What is meant with personality?

Line 450: In line with being careful about statements about influence, the authors should also be careful about statements about change. They would need a longitudinal design for such a statement. I suggest the authors rephrase and stick to conclusions that are possible based on the data available.

---

## [Author Response · Author response to Decision Letter 0]

25 Jan 2022

Responses to journal requirements

Response: We have gone through this guidance and amended the title pages and main body in accordance to these standard formatting guidelines. 

We note that the grant information you provided in the ‘Funding Information’ and ‘Financial Disclosure’ sections do not match. 

Response: We unable to identify the section on financial disclosure but can verify that the information provided in the funding information is correct. 

In your Data Availability statement, you have not specified where the minimal data set underlying the results described in your manuscript can be found. PLOS defines a study's minimal data set as the underlying data used to reach the conclusions drawn in the manuscript and any additional data required to replicate the reported study findings in their entirety. All PLOS journals require that the minimal data set be made fully available. We note that you have stated that you will provide repository information for your data at acceptance. Should your manuscript be accepted for publication, we will hold it until you provide the relevant accession numbers or DOIs necessary to access your data. If you wish to make changes to your Data Availability statement, please describe these changes in your cover letter and we will update your Data Availability statement to reflect the information you provide.

Response: Response: The minimum data related to this research is provided in the paper and in the supporting information. As mentioned in our data statement we wished to embargo the rest of the dataset until the paper was at the point of acceptance. However since it seems that this is not an option (based on feedback from the auditor) we have added the rest of the data to the Figshare website so that it is fully accessible. This includes all interview and group discussion transcripts. Observational data, visual data and data from handwashing demonstrations will not be made publicly available as it is unable to be fully anonymised. Therefore in our data statement we have indicated that some restrictions will apply and have provided more detail on this. The final dataset is available via this citation: White, Sian (2021): Interviews and group discussions with displaced populations in Iraq on the determinants of handwashing behaviour. figshare. Dataset. https://doi.org/10.6084/m9.figshare.17263829.v2. 

Please amend your list of authors on the manuscript to nsure that each author is linked to an affiliation. Authors’ affiliations should reflect the institution where the work was done (if authors moved subsequently, you can also list the new affiliation stating “current affiliation:….” as necessary).

Response: This has been amended in line with title page guidance. 

Please include captions for your Supporting Information files at the end of your manuscript, and update any in-text citations to match accordingly. 

Response: We have included this at the end of the manuscript. 

Reviewer 1 comments: 

The abstract reads well. However, the authors have indicated that the study took place in two countries (Irag and Kurdistan), as it appears in the title of their study. Yet the authors have only put Iraq in the key words. I suggest that they also add Kurdistan or remove Iraq from the key words. Or if its one country, ten that has to be made clear throughout the study. It is confusing that Kurdistan appears as a country in some other instance but it also appears as part of Iraq. This has to be clarified. 

Page 3, line 69; authors have mentioned that the setting of the study is in Northern Iraq. Yet in the title authors talk of two settings, thus Iraq and Kurdistan. Furthermore, in the abstract, authors have just mentioned of Iraq, and not Northern Iraq as is the case in the introduction.

I suggest that author need to be consistent I writing Iraq versus Northern Iraq. And also consistency on the setting of the study.

Response: Thank you for flagging this inconsistency. We have opted to use the term Northern Iraq throughout, including changing this in the title of the study. In the study site description, we have left in information about the specific study site locations, two of which fall within the Kurdistan Region of Iraq (not formally recognised as an autonomous state) and one which is located across the ‘border’ in the Ninewa governorate of Iraq. 

Lines 67 to 68, authors have introduced holistic behavioural framework and a range of participatory methods. This has been introduced suddenly and left in suspense. I suggest that authors provide a little more information about the holistic behavioural framework in the introduction section. What it is and its relevance/applicability to this study

Page 5, lines 93-101; Authors are talking about the framework used in the study. They have also talked about data collection methods; this is recommendable. However, I suggest that author should first talk about the framework they have claimed to use as written in the introduction (the holistic behavioural framework). Then they should link it with the framework written in this section. There is a missing link between the framework the authors have written in introduction on lines 67-68 and the framework with the constructs written in this section (lines 93-101) 

Response: Rather than alluding to a behavioural framework suddenly at the end of the introduction we have removed this so that it is introduced in full in the methods instead. 

On page 5, line 100, authors have written (SM1 1) and this seem to appear for the first time. I suggest that authors write it in full the first time before writing the abbreviation

Response: We have changed this to be in line with PLOS One guidance which suggests that Supplementary Materials should be referred to in text as S1 Appendix, S2 Appendix etc. 

Authors need to indicate those who did the observations, in terms of their expertise, qualifications, how many they were, how they ensured consistency between/among observers (validity). If they are the same who are mentioned on page 7 under data collection, (line 136), then they should first appear here.

Response: All data was collected by three of the authors SW, DI and WKI). We have been more clear about this in the data collection section so to indicate that this applied to the observations as well. We have added details into the observation section and data collection section about quality checking. 

Line 135, it s indicated that the initials are removed for review. But then immediately, in the next sentence 136) the initials appear with their nationality. I suggest they put the initials since they are already appearing on the second line for consistency. Also just as I mentioned above, the ones mentioned here to have collected data, and if they are the one referred to have made the observations, then they must appear under observation on page 5 first to avoid the audience wondering who these observers or data collectors are, and just to find their initials later in the document.

Response: Leaving these initials in was an error. All initials have now been added back into the manuscript. As mentioned above we have added clarification that these individuals also led the observation. 

Results are well written and clear. There is a good logical flow of presentation of the results. 

The interpretation of results is sufficiently substantiated by data. There is coherence between qualitative data sources, collection, analysis and interpretation. 

Response: We thank the reviewer for their kind feedback on the results and discussion sections.

Observation in relation to socially acceptable behaviour. This is because participants were observed practicing hand hygiene, and they knew they were video recorded. This limitations must come out strongly in their study.

Response: In the methods section we have added information about how reactivity was reduced during observation. We have also been more specific about the limitations of observation within this section on page 25. 

Reviewer 2 comments: 

Displacement, post-conflict settings, crisis-affected settings are all mentioned but may imply different things. I suggest to explain more precisely whether you researched and/or have findings relevant for crises more generally or only when people are displaced, any form of displacement or only when displaced due to conflicts.

Response: Thank you for pointing out this inconsistency in terms. We have tried to address this throughout the manuscript by being more specific about the study setting when referring to other research and being more specific that our study was focused on post-conflict displacement settings and that these are the settings that are findings are applicable to, although some aspects may be more generalisable. 

The phrasing of “change over the course of displacement” suggests a longitudinal design but it seems the design was not longitudinal. I suggest to phrase more precisely to avoid readers having inadequate expectations. Also, it is not entirely clear how it co-occurs with the different settings or whether there is both the time-component and the setting component

Response: That is correct, the design was not longitudinal but rather within the same study period we studied 3 separate populations who were at different stages of displacement. We have tried to clarify this throughout the text and have modified the title accordingly. 

Research on habit formation suggests that disruptive events may help change behaviour. It might be worthwhile reflecting on what this may mean for the manuscripts context. Check out research by Bas Verplanken.

Response: We appreciate this suggestion and have included some of Verplanken’s research as part of the introduction. 

Methodology should be extended. See some specific comments on issues particularly regarding collection of “prior” data.

Response: We have added some content to the methods on this and have tried to clarify within the results section that references to prior behaviours are based on participant self-reflections. In the limitations we have included that this self-reflection may be biased and not provide an accurate indication of past actions. 

The results section is quite dense and it is not entirely clear what the comparison criteria were. See specific comments on the latter point. I encourage the authors to reflect on whether the results could be reported in a different way that eases the processing of the information and makes the comparisons clearer.

Response: We do recognise that the results is both long and dense however we feel this is reflective of this type of qualitative research which is designed to provide a deep dive into the contextual determinants of behaviour. We have structured the results section according to the broad categories of the BCD framework which we used to inform our research. In each section we have tried to describe findings in a consistent way by describing patterns in current and past behaviours and including mentions of any outliner perceptions. In the methods we have added content to be clearer about the comparisons made, particularly in relation to pre and post conflict. We have opted not to change the results further given that the other reviewer found this structure to be clear. 

Quite a lot of typos, including punctuation. Also some inconsistencies in phrasing or use of terms. Please check spelling and punctuation when revising.

Response: We have reviewed the full text and corrected typos or grammar issues where they were identified. 

Abstract: not really clear what is meant with type of setting and whether “three post-conflict settings” are these different types or again another setting. Maybe rephrase and explain a bit more what was researched where.

Response: We have added in that these were 3 different types of settings and described these briefly in the abstract. 

Line 52: maybe inform which determinants were under reported.

Response: We have added this information to the introduction. 

Lines 59 and 64 might profit from some additional references on determinants in crisis-settings (first two) and over-estimation (last one) Contzen, N., & Mosler, H. J. (2013). Impact of different promotional channels on handwashing behaviour in an emergency context: Haiti post-earthquake public health promotions and cholera response. Journal of Public Health (Berlin, Heidelberg), 21(6), 559-573. doi:10.1007/s10389-013-0577-4, Contzen, N., & Mosler, H. J. (2015). Identifying the psychological determinants of handwashing: results from two cross-sectional questionnaire studies in Haiti and Ethiopia. American Journal of Infection Control, 43(8), 826-832., Contzen, N., De Pasquale, S., & Mosler, H. J. (2015). Over-reporting in handwashing self-reports: potential explanatory factors and alternative measurements. PLoS One, 10(8), e0136445 (22 pp.). doi:10.1371/journal.pone.0136445

Response: Thank you for these suggestions we have actually already referenced the third paper as part of the introduction. It is now also referenced in the limitations section of the paper. We have not included the first reference given that this focuses on delivery channels for hygiene promotion rather than determinants. We have included the second paper as a reference within the introduction also. 

Line 88: IDPs is used without term being introduced.

Response: This has been corrected

Line 87-90: not clear whether IDPs are also Arab or Shabak or something else, whether they are part of the “residents” displaced and returned, or whether they were additional, whether they were housed in other people’s houses or had their own which were also damaged etc. Explain more precisely.

Response: We have added that the IDPs were ethnically similar to the residents (they were Arab or Shabak). In the text we note that these IDPs originally came from neighbouring villages and that the IDP population was in addition to the 134 families who were from these village. We have added detail about where displaced families resided. 

Line 97: what type of characteristic is meant here??

Response: Characteristics is defined in full in Supplementary Materials 1 along with all other determinants. However to make it easier to understand in text we have added that this refers to socio-demographic characteristics.

Line 106: any action or only actions relevant to handwashing?

Response: Observers noted down all action that took place. This was done so that handwashing could be understood within broader routines and served to disguise the behaviour of interest, reducing reactivity. This has been explained in more depth in the text in the methods section. 

Line 112 and 117: not clear what the participatory activities were. Even if there is an SM explaining in detail, maybe say 1-2 sentences in the text to give readers a basic understanding on what had been done.

Response: We have added short descriptions of these activities within the sections on interviews and group discussions. 

Methods section: IDIs and FGDs are really hard to read. Maybe use interviews as abbreviation of IDIs and group discussions as abbreviation of FGDs.

Response: We have changed this throughout

Table 3: for all dichotomous variables half of the information is redundant

Response: We have removed the lines for male and illiteracy to reduce this redundancy

Determinants are said to “influence” behaviour or have an “impact”. To make any statements about influence or impact, at least a longitudinal design or better an experimental design is needed. Based on the data available, statements such as “seem to support or hinder” or “are related with” are possible, but nothing more.

Response: Thank you for pointing this out we have addressed these language issues throughout and agree that what we are describing are apparent associational relationships only. We have removed all references to impact. In some sections we have left influence as the term used because we feel this also describes associational relationships. We have added content to the limitations section to reflect that our findings are only associational. 

Line 176: you refer to 12 determinants but in the rest of the paragraph it is not that clear what is a determinant and what is “other information” you provide along. Restructure and report more precisely.

Response: We are not exactly sure what the reviewer is referring to when they say ‘other information’. In this particular section the 12 determinants refers to those which seemed to have a similar influence on behaviour prior to and across all research sites. The remaining determinants 21 determinants are described in the remaining part of the results. This is clearly articulated on line 229. 

Line 179, 181-182: “prior to displacement“, “before nor after displacement“ suggests you asked respondents how it had been before and how it is now. This was not clear to me based on method section and raises some questions as collection of retrospective data has quite some limitations. Please explain in more detail in methods section.

Response: As previously described we have amended the methods to make the description of this aspect of our work clearer. In the limitations we have also reflected on the limitations of behavioural recall. 

Results on determinants unaffected and affected by displacement: it is not entirely clear to me whether the authors conclude on whether or not a determinant is unaffected by displacement based on respondents statements about “before” and “after” (see above comment) or based on differences between settings. This is in line with a previous comment that it is not entirely clear whether both a time- and a setting-component were researched and how they are disentangled. The same problem occurs again in section on determinants before and during displacement based on title not clear whether this is now another analysis or a summary. I strongly suggest the authors revise and describe more precisely. Including being more precise on what your comparison criteria are.

Response: As previously explained the analysis considered behaviour prior to and post conflict and compared findings across study sites. We have made edits to reflect this in the methods and results. 

Line 354: injunctive norm is introduced but then the example which is given (if you asked 100 people within their area whether they wash their hands with soap at critical times) is rather an example of either descriptive norms or maybe of socially desirable responding. Check what concept you really mean here and whether the example fits to it or not. Line 358: being afraid that one would be judged is an aspect of injunctive norm � see previous comment � section on norms seems to need some revision.

Response: We have amended the wording in this section accordingly. 

Line 367 and following: whether something is a challenge or not does not say anything about whether it is important or not. Something can be a challenge but not important to me, or not a challenge but very important to me � not clear what this paragraph really taps into and what the key message is and how it is based on any of the statement by participants. Same issue again in line 453 and following.

Response: In this activity participants were specifically asked about challenges and were not asked about importance. However challenges were ranked in relation to each other to understand the prioritisation of these challenges. We believe that this is clear in the text and the methods section as we don’t suggest that asking about challenges is the same as asking about importance. 

Table 4: some information not very clear or useful. E.g., Age (being a child or an older person) has a negative “effect”. What does that mean exactly? What is meant with personality?

Response: The table provides a summary of the information described in the text. In this case ‘children were thought to need parental support to practice handwashing and older people were perceived to face barriers to handwashing because of reduced mobility’. In relation to personality we explain: ‘. Participants perceived personality to be one of the strongest determinants of handwashing. Multiple participants explained that if you were hygienic before the crisis, then you would continue to be hygienic when displaced.’ In both cases we have modified the content in the table to reflect these patterns more clearly. 

Line 450: In line with being careful about statements about influence, the authors should also be careful about statements about change. They would need a longitudinal design for such a statement. I suggest the authors rephrase and stick to conclusions that are possible based on the data available.

Response: Thank you we have adjusted the wording in this section.

---

## [Editor Report · Decision Letter 1]

11 Feb 2022

How is hygiene behaviour affected by conflict and displacement? A qualitative case study in Northern Iraq

PONE-D-21-15601R1

Dear Dr. White,

We’re pleased to inform you that your manuscript has been judged scientifically suitable for publication and will be formally accepted for publication once it meets all outstanding technical requirements.

Kind regards,

Hannah Tappis, DrPH, MPH

Academic Editor

PLOS ONE
---

## [Editor Report · Acceptance letter]

17 Feb 2022

PONE-D-21-15601R1 

How is hygiene behaviour affected by conflict and displacement? A qualitative case study in Northern Iraq 

Dear Dr. White:

I'm pleased to inform you that your manuscript has been deemed suitable for publication in PLOS ONE. Congratulations! Your manuscript is now with our production department. 

Kind regards, 

on behalf of

Dr. Hannah Tappis 

Academic Editor

PLOS ONE